# Modification of PMMA Cements for Cranioplasty with Bioactive Glass and Copper Doped Tricalcium Phosphate Particles

**DOI:** 10.3390/polym12010037

**Published:** 2019-12-25

**Authors:** Teresa Russo, Roberto De Santis, Antonio Gloria, Katia Barbaro, Annalisa Altigeri, Inna V. Fadeeva, Julietta V. Rau

**Affiliations:** 1Institute of Polymers, Composites and Biomaterials, National Research Council of Italy, V.le J.F. Kennedy 54-Mostra d’Oltremare Pad. 20, 80125 Naples, Italy; teresa.russo@unina.it (T.R.); angloria@unina.it (A.G.); 2Istituto Zooprofilattico Sperimentale Lazio e Toscana “M. Aleandri”, Via Appia Nuova 1411, 00178 Rome, Italy; katia.barbaro@izslt.it (K.B.); annalisa.altigeri@izslt.it (A.A.); 3AA Baikov Institute of Metallurgy and Materials Science, Russian Academy of Sciences, Leninsky prospect 49, 119334 Moscow, Russia; fadeeva_inna@mail.ru; 4Istituto di Struttura della Materia, Consiglio Nazionale delle Ricerche (ISM-CNR), Via del Fosso del Cavaliere 100, 00133 Rome, Italy; giulietta.rau@ism.cnr.it

**Keywords:** cranioplasty, PMMA, biocomposites, bioactive particles, bending properties, compressive properties, MTT assay, bacterial strains

## Abstract

Cranioplasty represents the surgical repair of bone defects or deformities in the cranium arising from traumatic skull bone fracture, cranial bone deformities, bone cancer, and infections. The actual gold standard in surgery procedures for cranioplasty involves the use of biocompatible materials, and repair or regeneration of large cranial defects is particularly challenging from both a functional and aesthetic point of view. PMMA-based bone cement are the most widely biomaterials adopted in the field, with at least four different surgical approaches. Modifications for improving biological and mechanical functions of PMMA-based bone cement have been suggested. To this aim, the inclusion of antibiotics to prevent infection has been shown to provide a reduction of mechanical properties in bending. Therefore, the development of novel antibacterial active agents to overcome issues related to mechanical properties and bacterial resistance to antibiotics is still encouraged. In this context, mechanical, biological, and antibacterial feature against *P. aeruginosa* and *S. aureus* bacterial strains of surgical PMMA cement modified with BG and recently developed Cu-TCP bioactive particles have been highlighted.

## 1. Introduction

Cranioplasty is a common technique for repairing bone defects in the cranium arising from cranial bone deformities, traumatic skull bone fracture, bone cancer, and infections. Surgery involves the use of biocompatible materials, and repair or regeneration of large cranial defects is particularly challenging from both functional and aesthetic point of view [1,2,3]. Poly-methyl-methacrylate (PMMA) is the biomaterial which has been most widely adopted for cranioplasty, and in some instances PMMA showed better long-term outcomes compared to other approaches based on frozen autologous bone [2,4,5,6]. As PMMA for cranioplasty purposes is conceived, at least four surgical approaches can be distinguished: The first and simplest one is the in situ application and polymerization of PMMA, consisting of a single step procedure applied intra-operatively [7,8,9]; the second ex vivo approach uses a plaster impression taken over the cranial defect for realizing a mould into which the PMMA prostheses is realised [10,11]; the third ex vivo approach considers a 3D scan of the cranial defect for realizing, through additive manufacturing (AM), a mould into which the PMMA prostheses is formed [12,13,14]; the fourth ex vivo approach uses 3D imaging of the defect in conjunction with AM to directly manufacture the PMMA prosthesis [15,16].

Medical-grade PMMA filaments [15], and gentamicin doped filaments [17], processed through AM, have also been investigated for cranioplasty.

For the in situ forming approach, PMMA is provided to surgeons in the form of a liquid monomer component in conjunction with a solid powder phase made of PMMA and/or copolymers. After mixing the liquid and powder phases, the reactive malleable paste is placed onto the skull defect and it is shaped for achieving a smooth prosthesis conforming to the normal contours of the patient’s skull [8,18]. The radical polymerization reaction is promoted by benzoyl-peroxide and amines enclosed in the powder and liquid phases, respectively [19,20]. The use of cold saline solution irrigation or of a damp gauze in saline solution placed between the setting acrylic resin and dura tissue are suggested for reducing temperature levels due to the heat developed during polymerization [11,21]. The incorporation of phase-change particles into the PMMA matrix has been suggested to reduce temperature levels occurring through the polymerization process [22]. Finally, the PMMA prosthesis is held in place using titanium plates and screws [10].

From a mechanical point of view, once polymerized, properties of PMMA are in between those of spongy and cortical bone [23]. Heat developed during the exothermic reaction limits the direct intra-operative use of PMMA-based cements, especially if large volumes of material need to be used. To overcome this limitation, PMMA can be processed through the moulding approach [24,25,26]. The main advancement of PMMA-based bone cements relies on modifications for improving biological and mechanical functions. The loading of antibiotics (i.e., gentamicin) into PMMA has been suggested to prevent infection [21,27], but a reduction of mechanical properties has been observed in bending [20]. A decrease of the compressive yield stress of about 50% has been measured for gentamicin loaded bone cement at a concentration lower than 1 wt% [28,29].

PMMA-based nanocomposites, incorporating silver [30,31] and gold [32] nanoparticles into the polymeric matrix, have been designed for antimicrobial purposes. For improving osteoconductivity and biocompatibility, several PMMA-based composites have been investigated for cranioplasty. Bioactive glass (BG) [14,33,34] and hydroxyapatite (HA) [35] represent the most common type of particles adopted for functionalizing and reinforcing the acrylic cement. Furthermore, copper doped tricalcium phosphate cements (Cu-TCP) have shown an antibacterial effect against Gram-negative bacteria [36].

In cranioplasty, surgical PMMA showed a rate of graft infection higher than 10% [6,8]. Most infections can be ascribed to strains that are resistant to common antibiotic therapies. Therefore, in the last decade, research has been focused in the development of novel antibacterial active agents to overcome issues related to mechanical properties and bacterial resistance to antibiotics such as gentamicin. In fact, it is recognised that gentamicin reduces mechanical properties of PMMA-based bone cements, and the efficacy of this type of antibiotic relies on the gentamicin entrapped on the free surface of the applied cement. Moreover, the antibacterial efficiency is strictly connected to the initial burst release. For these reasons, novel antibacterial agents in the solid state represented a challenging alternative to gentamicin. Within this context, we investigated mechanical, biological, and antibacterial features of surgical PMMA cement modified with BG and recently developed Cu-TCP bioactive particles.

## 2. Materials and Methods

Modified PMMA-based bone cements incorporating BG or Cu-TCP particles were prepared and systematically characterized.

### 2.1. BG Particles

Bioactive glass (BG) granules were obtained through sol-gel synthesis as previously described [33]. BG granules were synthesized from an aqueous solution of tetraethyl orthosilicate (TEOS), P(OEt)_3_, Ca(NO_3_)_2_⋅4H_2_O, NaNO_3_, Mg(NO_3_)_2_⋅6H_2_O, KNO_3_, NH_4_F, La(NO_3_)_3_⋅6H_2_O, and Ta(OC_2_H_5_)_5_, balanced in stoichiometric amounts. The solution underwent hydrolysis and polycondensation to obtain the following composition: SiO_2_ [43.68], P_2_O_5_ [11.10], CaO [31.30], Na_2_O [4.53], MgO [2.78], CaF_2_ [4.92], K_2_O [0.19], La_2_O_3_ [0.50], and Ta_2_O_5_ [1.00]. To catalyze TEOS and P(OEt)_3_ hydrolysis, HNO_3_ (0.1 M) was utilized. All products were from Sigma-Aldrich, Merck, Darmstadt, Germany, and were used as received. The synthesis was carried out in a Teflon bottle. Reactants were added to the mixture one by one, under vigorous stirring, every 30 min. The sol was left at room temperature for 10 days and then placed in an oven at 70 °C for 72 h to obtain a gel. This gel was dried at 120 °C for 48 h, and stabilized at 700 °C (heating rate 5 °C/min, cooling rate 20 °C/min).

### 2.2. Cu-TCP Particles

Cu2^+^-substituted TCP powders were obtained using the precipitation technique as previously described [36]. Briefly, 0.5 mol/L solution of Ca(NO_3_)_2_ were mixed with 0.5 mol/L solution of Cu(NO_3_)_2_ and calculated amount of 0.5 mol/L (NH_4_)_2_HPO_4_ solution was added dropwise to the solution. The pH was kept at 6.5–6.9 level by the addition of ammonia solution. After 30 min, the precipitate was filtered, washed with distilled water, dried at 80 °C, and calcined at 900 °C to form the whitlockite structure.

### 2.3. Modified PMMA-Based Bone Cements

PMMA bone cement (Palacos, Heraeus, Wehrheim, Germany) was modified with bioactive glass or copper doped tricalcium phosphate particles. Particles were dispersed in the solid PMMA phase as previously described [32]. Briefly, bioactive particles were dispersed into the solid PMMA phase through ultrasonic dispersion. The following PMMA-based formulations (*w*/*w*) were obtained: PMMA/BG 97.5/2.5, PMMA/BG 95/5, PMMA/BG 92.5/7.5, PMMA/BG 90/10, PMMA/Cu-TCP 97.5/2.5, PMMA/Cu-TCP 95/5, PMMA/Cu-TCP 92.5/7.5, PMMA/Cu-TCP 90/10.

The liquid MMA phase was added and hand mixed to the solid phase, hence the paste was poured into prismatic and cylindrical moulds in order to obtain specimens suitable for mechanical and biological investigations. For each modified bone cement composition, the batch volume consisted of 10 g of PMMA.

### 2.4. Mechanical Properties in Bending

The three-point bending test was performed through the Instron universal materials testing system (Model 5566, Instron, High Wycombe, UK) equipped with a load cell of 1 kN. The span length was set at 18 mm and loading rate was 1 mm/min. PMMA, PMMA/BG, and PMMA/CuTCP composites were poured into prismatic Teflon moulds in order to obtain specimens suitable for the three-point bending test (Figure 1a). Five specimens for each PMMA, PMMA/BG, and PMMA/CuTCP composite formulation were stored in a dark environment at room temperature for one week before performing mechanical testing. The bending modulus and strength were determined according ASTM D790.

### 2.5. Compression Strength

The compression test on cylindrical specimens was performed through the Instron universal materials testing system (Model 5566, Instron, High Wycombe, UK) equipped with a load cell of 5 kN. The loading rate was 1 mm/min. PMMA, PMMA/BG, and PMMA/CuTCP composites were poured into cylindrical Teflon moulds in order to obtain specimens suitable for the compression test (Figure 1b). Five specimens for each PMMA, PMMA/BG, and PMMA/CuTCP composite formulation were stored in a dark environment at room temperature for one week before performing mechanical testing. A preload of 50 N was applied to each specimen for 60 s, hence the compressive test was carried out up to 50% of strain, and the compression strength was measured.

### 2.6. Cell Viability Assay

Bone marrow is taken from a two-year-old butchered horse. Bone marrow was collected in tubes containing sodium citrate (Vacumed). Mononuclear cells, including mesenchymal stem cells, were collected from the diluted bone marrow aspirate by density gradient centrifugation at 800 g for 10 min. The cell pellet was rinsed thrice with phosphate buffered saline (PBS, Invitrogen AG, Basel, The Switzerland), hence cells were re-suspended in an alpha-minimal essential medium (α-MEM, Gibco BRL, Life Technologies Limited, Inchinnan, UK) with 20% fetal calf serum (FCS, Gibco BRL, Life Technologies Limited, Inchinnan, UK), seeded into a flask of 75 cm^2^ (Corning, Oneonta, NY, USA) and expanded in an incubator at 5% CO_2_ at 37 °C. The MTT (Sigma-Aldrich, Merck, Darmstadt, Germany) tetrazolium salt colorimetric assay was performed to determine cytotoxicity and proliferation of cells cultured on PMMA and PMMA-based composites. Bone marrow mesenchimal stem cells (BMMSC) of passage 3 and 75–80% confluency were enzymatically detached and distributed at a concentration of 40,000 cells/mL into tissue culture plates, each consisting of 24 wells (Corning, Oneonta, NY, USA). Culture plates were incubated for 24 h at 5% CO_2_ at 37 °C. Hence, PMMA, PMMA/BG, and PMMA/CuTCP- based composites were layered into the wells. For each PMMA-based biocomposite, the MTT assay was performed in triplicate. BMMSCs growth and viability was evaluated after 24 h of incubation. The culture medium was removed from each well, replaced by a solution of 0.3 mL consisting of MTT 0.5 mg/mL in α-MEM, and incubated for 3 h at 5% CO_2_ at 37 °C. Hence, the solution of MTT in α-MEM was removed, replaced with 1.5 mL isopropanol, and incubated for 1 h. Finally, the concentration of formazan was quantified by optical density measurement at 600 nm through the BioPhotometer (Eppendorf AG, Hamburg, Germany).

### 2.7. Cell Differentiation Assay

BMMSCs capability to differentiate into chondrogenic lineage was evaluated for all PMMA and PMMA composites formulations. BMMSCs of passage 3 and 75–80% confluency were enzymatically detached and distributed at a concentration of 40,000 cells/mL into tissue culture plates, each consisting of 24 wells. Culture plates were incubated for 24 h at 5% CO_2_ at 37 °C. Hence, PMMA, PMMA/BG, and PMMA/CuTCP-based composites were layered into the wells. After 24 h of incubation, the cell monolayer was stimulated toward the chondrogenic differentiation for three weeks by using α-MEM media supplemented with 1% of FCS supplemented with 0.1 µM dexamethasone (Sigma-Aldrich, Merck, Darmstadt, Germany), 6.25 μg/mL insulin (Sigma-Aldrich, Merck, Darmstadt, Germany), 50 nM ascorbic acid (Sigma-Aldrich, Merck, Darmstadt, Germany), and 10 ng/mL TGFα (Sigma-Aldrich, Merck, Darmstadt, Germany). The negative control was obtained by using α-MEM containing 1% of FCS. BMMSCs differentiation was evaluated through the Alcian Blue colorimetric assay (Sigma-Aldrich, Merck, Darmstadt, Germany) by incubating each sample into a 10% *v*/*v* formalin solution for 1 h at room temperature, washing, and rinsing in distilled water, staining for 15 min at room temperature with a solution of alcian blue in acetic acid (1% alcian blue into a 3% acetic acid solution, pH = 2.5), and finally washing thrice with 3% acetic acid solution. Intra and extra cell blue stained glycosaminoglycans were observed through inverted optical microscopy (Nikon Eclipse TE2000-U, Iowa City, USA). For each PMMA-based biocomposite, the cell differentiation assay was performed in triplicate.

### 2.8. Viability of Bacterial Strains

*P. aeruginosa*, *S. aureus*, and *E. coli* were used. Each bacterial strain was grown in a brain heart infusion (BHI, Sigma-Aldrich, Merck, Darmstadt, Germany) for 24 h at 37 °C by shaking at 250 rpm. For each PMMA, PMMA/BG, and PMMA/CuTCP composites, three samples were used, each kept for 72 h in BHI. Hence, 10 µL of bacterial suspensions, collected from the overnight growth, were transferred to each well, and anti-bacterial activity of each PMMA-based substrate was investigated following incubation at 37 °C for 24 h. For each PMMA-based biocomposite the test was performed in triplicate, and optical density at 600 nm through the BioPhotometer (Eppendorf AG, Hamburg, Germany) was measured.

### 2.9. Statistical Analysis

All mechanical and biological measurements were evaluated through multi-way analysis of variance, and differences observed between the means of PMMA, PMMA/BG, and PMMA/CuTCP composites, were considered significant at *p* < 0.05.

## 3. Results

### 3.1. Mechanical Properties in Bending

PMMA and PMMA-based composites have shown a mechanical behaviour in bending typical of brittle materials: The stress linearly increases as the strain is increased, and a brittle fracture occurs at strain values of about 3% (Figure 1a). Bending modulus and strength of PMMA, PMMA/BG, and PMMA/CuTCP composites are reported in Figure 2a,b, respectively. Low amounts (i.e., 2.5%_w_) of CuTCP particles significantly increased the bending modulus of PMMA (*p* < 0.05). However, at low concentration of CuTCP particles, the bending strength increase is not significant. In addition, BG particles have shown a slight increase of the bending modulus, but this increase is not statistically significant. For both BG and CuTCP particles, a decrease of bending properties (i.e., modulus and strength) is observed as the amount of particles is increased. In particular, only the bending modulus and strength of the PMMA/BG 90/10 are significantly lower (*p* < 0.05) than those of plain PMMA.

### 3.2. Compression Strength

PMMA and PMMA-based composites have shown a mechanical behaviour in compression typical of ductile materials: The stress linearly increases as the strain is increased up to the yielding point and an extended stress plateau is observed (Figure 1b). No specimen underwent failure up to a deformation of 50%. Compressive strength (i.e., stress plateau level) of PMMA, PMMA/BG, and PMMA/CuTCP composites are reported in Figure 2c. At any of the investigated amounts of CuTCP, the compressive strength is not different from that of the plain PMMA cement. A similar result has been observed for BG particles, only for the PMMA/BG 90/10 a significant decrease has been observed (*p* < 0.05). However, with the exception of the PMMA/BG 90/10 composite, all bioactive formulations show a compressive strength higher than the limit value of 70 MPa recommended by ISO 5833.

### 3.3. Cell Viability Assay

Figure 3 reports cell viability results according to the MTT assay for the different PMMA-based biocements. Optical density values at 600 nm, suggest that PMMA, PMMA/BG, and PMMA/CuTCP composites induce no significant inhibition of BMMSCs growth occurs.

### 3.4. Cell Differentiation Assay

Figure 4 shows the capability of BMMSCs to differentiate, after three weeks of incubation, into chondrogenic lineage on PMMA, PMMA/BG, and PMMA/CuTCP composites, as suggested by alcian blue staining.

Plain PMMA supplemented with a chondrogenic medium (Figure 4a), BMMSCs supplemented with a condrogenic medium (positive cell control, Figure 4b), and BMMSCs (negative cell control, Figure 4c) were also used as controls.

### 3.5. Viability of Bacterial Strains

Figure 5 shows bacterial growth on PMMA, PMMA/BG, and PMMA/CuTCP composites. Results suggest that modification of bone cements with BG and CuTCP particles are effective in reducing *P. aeruginosa* growth (Figure 5a). In particular 2.5%_w_ of BG or CuTCP particles already produce a significant reduction of *P. aeruginosa* growth (*p* < 0.05). However, 5%_w_ of BG or CuTCP particles reduces by about 80% and 75% *P. aeruginosa* growth, respectively. Modification of bone cements with BG and CuTCP particles are effective in reducing *S. aureus* growth (Figure 5b). In particular, at a concentration of 5 wt% of BG or CuTCP, a significant reduction (*p* < 0.05) of *S. aureus* growth is observed. No significant inhibition of *E. coli* growth has been observed (Figure 5c).

## 4. Discussion

Over the past decade, a wide range of polymer-based composites have been investigated to repair cranial bone, and PMMA-based bone cements has been most widely adopted for cranioplasty [7,8,9]. However, surgical PMMA showed a rate of graft infection higher than 10% [6,8].

Modification of PMMA cements with bioactive particles have been suggested to prevent infections [14,30,31,32].

BG and HA represent the most common type of particles adopted for modifying bone cements [14,35] and we investigated the effects of BG and Cu-TCP particles on mechanical properties, biological behaviour, and antibacterial capability.

The brittle behaviour observed through bending tests of PMMA and PMMA-based composites, as well as bending modulus and strength (Figure 2a,b), is consistent with literature data reported for the Palamed bone cement [37]. On the other hand, the ductile behaviour in compression observed for PMMA and compression strength measurements (Figure 1b and Figure 2c) are consistent with literature data reported for the stress vs. strain curves and strength of Palamed bone cement [28,29]. 2.5 wt % of CuTCP particles significantly increased the bending modulus of PMMA (*p* < 0.05). However, the bending modulus decreased as the amount of CuTCP particles is increased. A similar trend has been observed in bending for PMMA/BG composites (Figure 2). No significant effect of CuTCP particles on compression strength has been observed for PMMA/CuTCP composites, while a slight but significant decrease (*p* < 0.05) of the compression strength has been observed for PMMA/BG 90/10_w_ composites (Figure 2c). With the exception of the PMMA/BG 90/10 composite, all bioactive formulations show a compressive strength higher than 70 MPa, thus satisfying the recommendation of ISO 5833. These mechanical results suggest that both BG and CuTCP particles provide a reinforcement effect for the PMMA matrix only at low amount (i.e., 2.5 wt%). The negligible reinforcement effect of the investigated particles at high concentration (>5 wt%) may be ascribed to a clustering effect of particles and to a weak particle-polymer matrix interface. Further research is needed to improve particles dispersion and particle-polymer matrix interface.

From a biological point of view, PMMA, PMMA/BG, and PMMA/CuTCP composites have shown to be non-toxic, as the MTT assay suggests that the investigated substrates induce no significant inhibition of BMMSCs growth (Figure 3). This result is corroborated by the capability of BMMSCs to differentiate, after three weeks of incubation, into chondrogenic lineage on PMMA, PMMA/BG, and PMMA/CuTCP composites (Figure 4).

Most infections related to PMMA-based bone cements for cranyoplasty can be ascribed to strains that are resistant to common antibiotic therapies. Therefore, in the last decade, a huge research has been focused on the development of novel antibacterial active agents to overcome bacteria resistance to antibiotics [14,30,31,32]. Both BG and CuTCP particles have shown to be an interesting candidate for the modification of surgical bone cements. In particular, BG and CuTCP particles are effective in reducing *P. aeruginosa* growth (Figure 5a). At a concentration of 2.5 wt%, BG or CuTCP particles already produce a significant reduction of *P. aeruginosa* growth (*p* < 0.05). However, 5 wt% of BG or CuTCP particles reduces by about 80% and 75% *P. aeruginosa* growth, respectively (Figure 5a). *S. aureus* is the most common pathogen affecting cranioplasty [38]. Modification of bone cements with BG and CuTCP particles have shown to be also effective in reducing *S. aureus* growth (Figure 5b). In particular, at a concentration of 5%_w_ of BG or CuTCP, a significant reduction (*p* < 0.05) of about 50% of *S. aureus* growth is observed. This result is consistent with literature data reported for growth reduction of S. epidermidis strains onto PMMA cements modified with copper doped BG particles [34]. No significant inhibition of E. coli growth has been observed (Figure 5c). It is reported that *E. coli* strains can survive in copper-rich environments as they possess plasmid-encoded genes that confer copper resistance [39]. Accordingly, antibacterial results suggest that BG and CuTCP particles can be considered as promising candidates for modifying PMMA-based bone cements. However, further research is needed to further improve activity toward bacterial strains, such as *E. coli*. 

A number of factors may be involved in the mechanism of bacterial growth inhibition or bacterial killing [40,41]. One of the key mechanism causing cell damage has been termed contact killing. This mechanism consists of the following steps: Copper dissolution from the external surface, ruptures of cell membrane, loss of cytoplasmic content and membrane potential, production of reactive oxygen species, and degradation of genomic and plasmid DNA [40,41].

## 5. Conclusions

Within the limitations of this investigation it can be concluded that the incorporation of bioactive particles (i.e., CuTCP and Bioactive glass particles) into PMMA-based bone cements do not alter mechanical properties in bending and compression. With the exception of the PMMA/BG 90/10 composite, all bioactive formulations show a compressive strength higher than 70 MPa, thus satisfying the recommendation of ISO 5833. PMMA/BG and PMMA/CuTCP biocomposites are nontoxic and they show antibacterial activity against *P. aeruginosa* and *S. aureus* bacterial strains. However, further research is needed to further improve activity of PMMA-based bone cements toward bacterial strains such as *E. coli*.

## Figures and Tables

**Figure 1 polymers-12-00037-f001:**
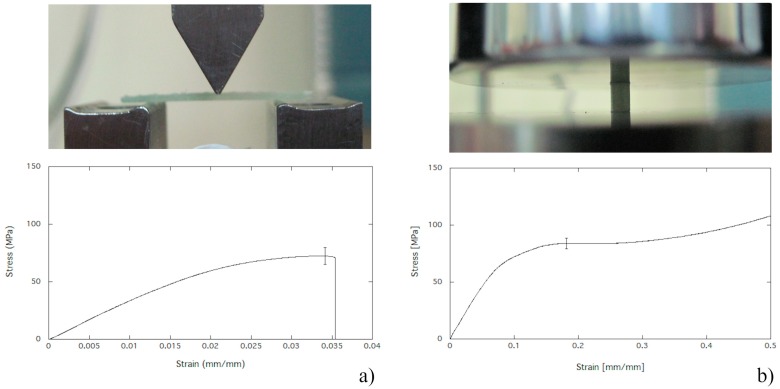
Mechanical testing setup and typical stress strain behaviour in: (**a**) Three point bending; (**b**) compression.

**Figure 2 polymers-12-00037-f002:**
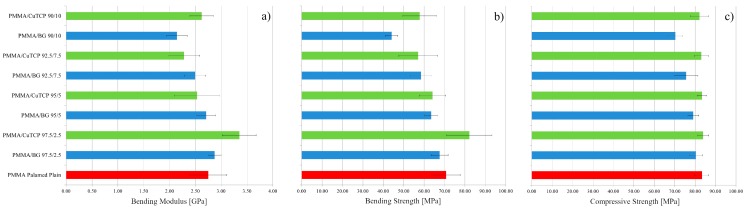
Mechanical properties of PMMA, PMMA/BG and PMMA/CuTCP composites. (**a**) Bending modulus; (**b**) bending strength; (**c**) compression strength.

**Figure 3 polymers-12-00037-f003:**
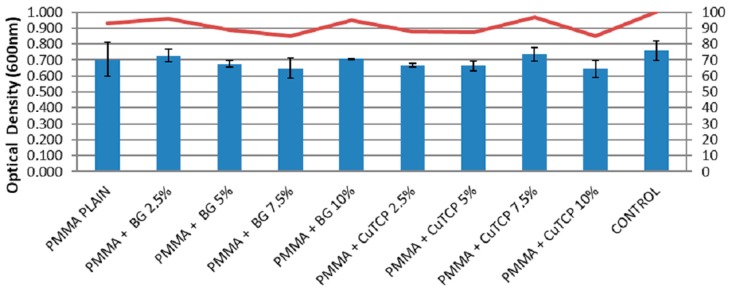
MTT assay. Optical density mean values at 600 nm and standard deviation of BMMSCs grown on the different modified PMMA based bone cement.

**Figure 4 polymers-12-00037-f004:**
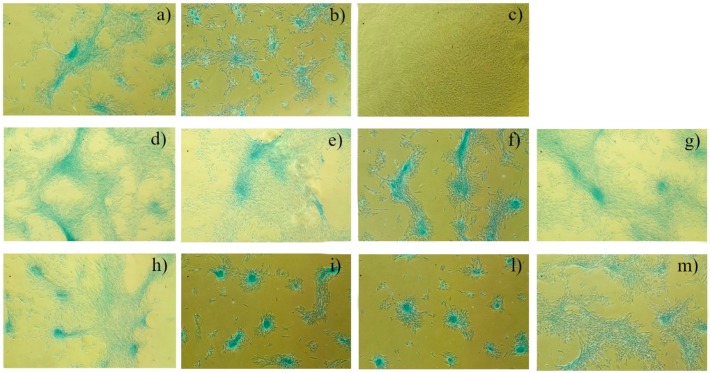
Capability of BMMSCs to differentiate, after three weeks of incubation, into chondrogenic lineage. (**a**) PMMA plain supplemented with chondrogenic medium; (**b**) positive cell control (with chondrogenic medium, no PMMA); (**c**) negative cell control (neither PMMA no chondrogenic medium); (**d**) PMMA/BG 97.5/2.5; (**e**) PMMA/BG 95/5; (**f**) PMMA/BG 92.5/7.5; (**g**) PMMA/BG 90/10; (**h**) PMMA/CuTCP 97.5/2.5; (**i**) PMMA/CuTCP 95/5; (**l**) PMMA/CuTCP 92.5/7.5; (**m**) PMMA/CuTCP 90/10.

**Figure 5 polymers-12-00037-f005:**
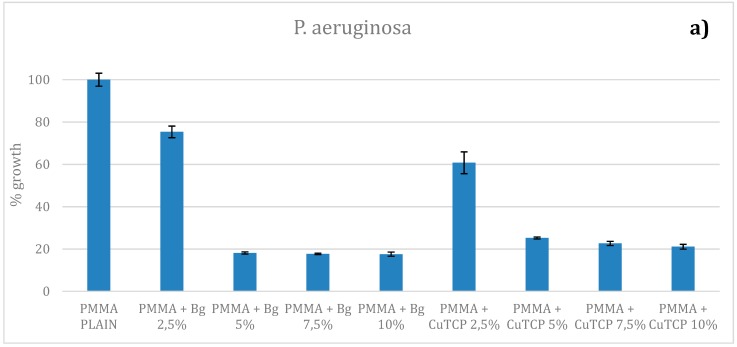
Antibacterial capability of PMMA, PMMA/BG, and PMMA/CuTCP composites. (**a**) *P. aeruginosa* growth; (**b**) *S. aureus* growth; (**c**) *E. coli* growth. Standard deviation and statistical analysis.

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
