# Peer review of "Modification of PMMA Cements for Cranioplasty with Bioactive Glass and Copper Doped Tricalcium Phosphate Particles"

_polymers, 2019, doi:10.3390/polym12010037_

Round 1
Reviewer 1 Report
Section 1.5. Compression strength:
Figure 1 ((a) and (b)) - better resolution is required.
Section 3.1. Mechanical properties in bending:
Authors claim that "Low amounts (i.e. 2.5%w) of CuTCP particles significantly increased the bending modulus of PMMA (p<0.05). However, at low concentration of CuTCP particles, the bending strength increase is not significant.". It is not according with results presented in Figure 2a). A clear Figure is required comprising a right position between the names of PMMA composites and the corresponding graphical bars. Please, provide a description accordingly.
Section 4. Discussion
Again, authors claim that "2.5wt.% of CuTCP particles significantly increased the bending modulus of PMMA (p<0.05). However, the bending modulus decreased as the amount of CuTCP particles is increased.".... "These mechanical results suggest that both BG and CuTCP particles provide a reinforcement effect for the PMMA matrix only at low amount (i.e. 2.5wt.%).". Revisions are required according to the recommendations above mentioned (section 3.1).
Author Response
We thank reviewer #1 for his precious suggestions. The following are the corrections that we applied.
Section 1.5. Compression strength:Figure 1 ((a) and (b)) - better resolution is required.
We increased the resolution of Figure 1.
Section 3.1. Mechanical properties in bending:Authors claim that "Low amounts (i.e. 2.5%w) of CuTCP particles significantly increased the bending modulus of PMMA (p<0.05). However, at low concentration of CuTCP particles, the bending strength increase is not significant.". It is not according with results presented in Figure 2a). A clear Figure is required comprising a right position between the names of PMMA composites and the corresponding graphical bars. Please, provide a description accordingly.
The reviewer is perfectly wright. We are in debt with him as we did not realize that there was a mistake in the labels of the y axis. Accordingly we modified the y axis labels. We also aligned the y axis labels with the bars of the plot, and we increased the resolution of Fig. 2.
Section 4. DiscussionAgain, authors claim that "2.5wt.% of CuTCP particles significantly increased the bending modulus of PMMA (p<0.05). However, the bending modulus decreased as the amount of CuTCP particles is increased.".... "These mechanical results suggest that both BG and CuTCP particles provide a reinforcement effect for the PMMA matrix only at low amount (i.e. 2.5wt.%).". Revisions are required according to the recommendations above mentioned (section 3.1).
As previously reported, there was a mistake in Fig. 2, which has now been corrected.
Reviewer 2 Report
The article ‘Modification of PMMA cements for cranioplasty with 2 bioactive glass and copper doped tricalcium 3 phosphate particles’ describes the synthesis of polymer based cement material with bioactive occlusions. This manuscript is nicely drafted, also providing a good overview of background literature. The conclusions drawn are well supported by experimental data.
Minor points of improvement are listed below.
1. Introduction: It would be a nice addition to describe the key knowledge gap which is addressed. The literature description is comprehensive. Please describe the key aspects or novelty of the described work in context.
2. Cu-TCP particles : What was the batch volume applied during synthesis?
3. Line 188: Escherichia
4. A copper based TCP particles are shown to inhibit the growth of P. aeruginosa and S. aureus. Does the mechanism involve the leaching of the copper ions from the material or does the substrate itself suppress bacterial growth? A brief discussion on this aspect will be interesting for the field.
Please see:
Grass, Gregor, Christopher Rensing, and Marc Solioz. "Metallic copper as an antimicrobial surface." Appl. Environ. Microbiol. 77.5 (2011): 1541-1547.
Santo, Christophe Espírito, et al. "Bacterial killing by dry metallic copper surfaces." Appl. Environ. Microbiol. 77.3 (2011): 794-802.
Author Response
We thank reviewer #2 for his nice comments on the manuscript and for his important suggestions. The following are the corrections that we applied.
Minor points of improvement are listed below.
Introduction: It would be a nice addition to describe the key knowledge gap which is addressed. The literature description is comprehensive. Please describe the key aspects or novelty of the described work in context.
The following phrase was added in the Introduction section:
In cranioplasty, surgical PMMA showed a rate of graft infection higher than 10% [6,8]. Most infections can be ascribed to strains that are resistant to common antibiotic therapies. Therefore, in the last decade, research has been focused in the development of novel antibacterial active agents to overcome issues related to mechanical properties and bacterial resistance to antibiotics such as gentamicin. In fact, it is recognised that gentamicin reduces mechanical properties of PMMA based bone cements, and the efficacy of this type of antibiotic relies on the gentamicin entrapped on the free surface of the applied cement. Moreover, the antibacterial efficiency is strictly connected to the initial burst release. For these reasons, novel antibacterial agents in the solid state represented a challenging alternative to gentamicin. Within this context, we investigated mechanical, biological and antibacterial feature of surgical PMMA cement modified with BG and recently developed Cu-TCP bioactive particles.
Cu-TCP particles : What was the batch volume applied during synthesis?
For each modified bone cement composition, the batch volume consisted of 10g of PMMA.
Line 188: Escherichia
We revised the term in the manuscript, and it is highlighted in yellow.
A copper based TCP particles are shown to inhibit the growth of P. aeruginosa and S. aureus. Does the mechanism involve the leaching of the copper ions from the material or does the substrate itself suppress bacterial growth? A brief discussion on this aspect will be interesting for the field.
We thank the reviewer for giving us the opportunity to speculate on the mechanism involved in bacterial killing. We appreciated the suggested references that have been integrated in our manuscript. The following paragraph was added in the Discussions section.
A number of factors may be involved in the mechanism of bacterial growth inhibition or bacterial killing [40,41]. One of the key mechanism causing cell damage has been termed contact killing. This mechanism consists of the following steps: copper dissolution from the external surface, ruptures of cell membrane, loss of cytoplasmic content and membrane potential, production of reactive oxygen species, and degradation of genomic and plasmid DNA [40,41].